# Factors Affecting Cultural Competence in a Sample of Nursing Students during the Prolonged COVID-19 Pandemic in Republic of Korea: A Cross-Sectional Study

**DOI:** 10.3390/ijerph192215181

**Published:** 2022-11-17

**Authors:** Hyeran An, Juhyun Jin, Taehyun Kim

**Affiliations:** 1Institute of Nursing Science, College of Nursing, Daegu Catholic University, Daegu 42472, Republic of Korea; 2Department of Nursing, Yeungnam University College, Daegu 42415, Republic of Korea

**Keywords:** cultural competence, cultural intelligence, ethnocentrism, global competence, nursing student

## Abstract

Globally, foreign citizens, particularly ethnic and racial minorities, experienced discrimination and received imbalanced medical services and insufficient economic resources during the COVID-19 pandemic. This study aimed to examine the factors that affect the cultural competence of nursing students. This is descriptive cross-sectional study adheres to Strengthening the Reporting of Observational studies in Epidemiology (STROBE) guidelines. A convenience sample of 235 nursing students from two nursing colleges in D city completed an online Google Forms questionnaire from 9 August to 12 August 2022. The self-report questionnaire included a sociodemographic data form, a cultural intelligence scale, an ethnocentrism scale, a global competence scale, and a cultural competence scale. The mean score of cultural competence was 95.39 ± 15.64 (out of 135 points); cultural competence was significantly positively correlated with cultural intelligence and global competence (*p* < 0.001), and significantly negatively correlated with ethnocentrism (*p* < 0.001). The factors that significantly affected cultural competence were cultural intelligence (β = 0.31, *p* < 0.001) and global competence (β = 0.37, *p* < 0.001). The explanatory power of these effects was 47.3%. To improve the cultural competence of nursing students, it is necessary to develop, apply, and evaluate the results of curriculum and programs that can enhance the cultural intelligence and global competence of nursing students.

## 1. Introduction

Recent reports of the COVID-19 outbreak showed that vulnerable population such as racial and ethnic minorities, including pregnant women, infants, and older adults, were at a higher risk of severe morbidity, complications, and death from the virus [1]. Globally, foreign citizens, particularly of Asian origin, have experienced discrimination and hostility related to the COVID-19 pandemic, and have received imbalanced medical services and insufficient economic resources [2]. In the Republic of Korea, foreign citizens have also experienced discrimination toward vaccination, national subsidy, and distribution of masks [2]. Even before the outbreak of COVID-19, this global health issue involving foreign citizens already existed, but currently, more attention should be placed on the provision of health services to this vulnerable population [3]. Because these issues are not limited to the current COVID-19 pandemic, when the pandemic is over, it will become more important; moreover, as more people participate in the global health network, there may be conflicts of interest among individuals [4].

Therefore, in time of crisis, healthcare professionals with a high cultural competence are needed, even more than before [5], particularly frontline healthcare workers, such as nurses. Because a high cultural competence can increase the quality of the interaction developed between patients and nurses and increase the provision of safe, equal, and appropriate health services [6], nurses’ cultural competence should be developed through training continually, starting in undergraduate school [7]. Furthermore, as future professional nurses, nursing students also need high cultural competence to provide tailored nursing care that considers patients’ cultural characteristics in hospitals, schools, and communities, through specialized education and practice [8,9,10,11,12]. However, in the Republic of Korea, 21.9–28.7% of nursing students had foreign friends, 11.6–36.4% had previously stayed in foreign countries for more than a month, and 4.6–33.5% had experience of international health care [4,9,13]. These low multicultural contact experiences can negatively affect nurses’ cultural awareness and knowledge [14]. Even nursing colleges in the Republic of Korea have organized and operated multicultural courses as part of the “competency and awareness of changes in domestic and foreign healthcare policies” presented in the program learning outcomes of the Korea Accreditation Board of Nursing Education. However, it has been reported that current cultural education in nursing curriculums is not enough to improve students’ cultural competence [10,14]. Furthermore, under the prolonged COVID-19 pandemic, physical and environmental isolation measures, such as social distancing inside and outside the country, led to mental and social isolation, including an insufficient interaction among individuals that also affected the quality of the health care services provided [4,5,15]. Consequently, the cultural competence of nursing students and nurses has reportedly decreased since the COVID-19 pandemic [4,5,9,16]. However, few studies have focused on the cultural competence of nursing students during the COVID-19 pandemic in the Republic of Korea. Therefore, it is necessary to examine the cultural competence of nursing students and identify the factors that influence it in order to develop a cultural program that improves the provision of culturally appropriate health care services after the COVID-19 pandemic. Previous studies conducted before the onset of the COVID-19 pandemic have found that factors such as being exposed to multicultural experiences [8,9], providing nursing care to foreign patients [17], multicultural education [8,9,17,18], English and communication skills [8,17,18], cultural empathy [8,14], multicultural awareness [14], global citizenship [13], global health [4], self-efficacy [18], global competence [19], and cultural intelligence [20] had an influence on cultural competence of nursing student. in addition, a cultural competence measurement tool for Korean nursing students was developed [12]. Cultural intelligence, ethnocentrism, and global competence are considered to be meaningful factors for increasing the cultural competence among nursing college students. Cultural intelligence is an individual’s ability to effectively act and manage in various cultural contexts [21], and is an essential factor to enhance cultural sensitivity and competence [20,21]. Until now, previous studies have focused on the relationship between cultural competence and emotional intelligence, empathy, and concepts similar to cultural intelligence, and few studies could confirm the relationship between cultural intelligence and competence [22].

In contrast, ethnocentrism refers to evaluating other people’s groups or cultures based on one’s own culture, without accepting other cultures. It is associated with a sense of cultural superiority, and is a concept dissimilar to cultural relativism or acceptability [23]. High ethnocentrism in health professionals demean the diverse cultures of patients and collaborators, thereby impairing their ability to provide culturally appropriate care [7]. A study showed that there was a significant correlation between the ethnocentrism and cultural sensitivity of nursing students in Turkey and Taiwan [24,25]. In the Republic of Korea, a study investigated the negative relationship between the ethnocentrism and cultural competence of clinical nurses [16].

Global competence refers to multicultural communication capabilities, such as cultural knowledge, language, global dynamics, understanding, and openness to the external environment and others, multicultural awareness, and foreign language proficiency [26]. Although it has been emphasized as a prerequisite for improving cultural competencies, it is rare to find a study that focus on the relationship between global competence and cultural competence [19]. Nursing students with high cultural intelligence, global competence, and ethnocentrism are expected to be have higher cultural competencies. Under the prolonged COVID-19 pandemic, it is important to adopt an integrative approach that identifies the factors that influence cultural competence. Furthermore, other aspects such as number of trips abroad, number of foreign friends, interest in news or information about foreign countries, preference for foreign films, dramas and documentaries, and experience providing nursing care to foreign patients. 

Therefore, this study aimed to examine the cultural intelligence, ethnocentrism, global competence, and cultural competence level of Korean nursing students and the relationship between them, and identify the factors that affect cultural competence. Based on the results, we intend to provide primary data essential for planning subjects and online and offline non-curricular programs to enhance the cultural competences of nursing students after the COVID-19 pandemic.

## 2. Materials and Methods

### 2.1. Study Design

This is cross-sectional descriptive study designed to identify the factors affecting the cultural competence of nursing students the Prolonged COVID-19 pandemic in the Republic of Korea. This is following the STROBE (Strengthening the Reporting of Observational studies in Epidemiology) guidelines for cross-sectional studies. 

### 2.2. Participants

A convenience sample of 240 nursing students registered in two College of Nursing in metropolitan D city. Participants understood the purpose of this study and voluntarily agreed to participate, and students on leave were excluded from the study. The sample number was calculated using a G-power 3.1.9.4, where the minimum number required for a significance level of 0.05, power of 0.95, the effect size of 0.15, and 13 independent variables was 189. Data from 240 people were collected considering approximately 20% rate of attrition. Of these, five uncompleted questionnaires were excluded, and data from 235 people were used for the final analysis. 

### 2.3. Data Collection 

Data were collected from 9 August 2022 to 12 August 2022, and all the participants were recruited via digital social networks and added to a Kakao Talk (mobile messaging app by Kakao Corporation in the Republic of Korea) group chat according to their specific grade, using an online Google Form questionnaire. First, the researchers contacted the representatives of the first to fourth graders of the two nursing universities that allowed guidance and related matters of this study through phone calls and posted them in each chat room group so that participants who participated voluntarily have easy access. They received consent information on the online questionnaire before the survey began about the study’s aims, the autonomy of participation, and confidentiality of the collected data for research purposes only. If the participant did not agree, the online link automatically terminated.

### 2.4. Instruments 

#### 2.4.1. Sociodemographic Characteristics Questionnaires 

The researchers developed a sociodemographic characteristics questionnaire to measure the following nine items: age, gender, grades, clinical practice experience, number of trips abroad, number of foreign friends, interest in news or information about foreign countries, preference for foreign films, dramas, and documentaries, and experience of providing nursing care to foreign patients.

#### 2.4.2. Cultural Intelligence Scale (CIS)

Cultural intelligence was measured by Cultural Intelligence Scale (CIS) that was developed by Ang et al. [21]. The validity and reliability of Korean version of CIS were establish by Baek and Chang [27]. It comprises 20 items in four sub-dimensions: cognitive (6 items), meta-cognitive (4 items), behavioral (5 items), and motivational (5 items), with the scoring of a seven-point Likert scale. The minimum score is 20 and the maximum is 140, with higher scores indicating a higher level of cultural intelligence. The scores of 20–54 denote “low cultural intelligence”, 55–104 “moderate cultural intelligence”, and 105–140 “high cultural intelligence”. Cronbach’s α was 0.86 in the original version [21], 0.93 for the Korean version [27], and 0.92 in this study.

#### 2.4.3. Generalized Ethnocentrism Scale (GENE)

Ethnocentrism was measured by Generalized Ethnocentrism Scale (GENE) that was developed by Neuliep [28]. The validity and reliability of Korean version of GENE was establish by Ahn [16]. In this study, 15 items of the 22 scale items were used for data collection based on the Ahn’s study [16]. Questions were rated on a five-point Likert scale, ranging from 1 (strongly disagree) to 5 (strongly agree). Higher scores indicate higher ethnocentrism and over 55 as “high Ethnocentrism”. Cronbach’s α was 0.80 in the original version [28], 0.83 for the Korean version [16], and 0.83 in this study.

#### 2.4.4. Global Competence Scale (GCS) 

Global competence was measured by Global Competence Scale that was developed for Korean university students by Shin and Noh [29]. It comprises 27 items in five sub-dimensions: English proficiency (12 items), creative thinking (4 items), self-expression capability (4 items), cultural flexibility (4 items), and global activities (3 items), with the scoring of a five-point Likert scale. The minimum score is 27 and the maximum is 135, with higher scores indicating a higher level of global competence. Cronbach’s α was 0.92 in the original version [29] and 0.89 in this study.

#### 2.4.5. Cultural Competence Scale (CCS)

Cultural Competence was measured by Cultural Competence Scale (CCS) that was developed for Korean nursing students by Han and Chung [12]. It comprises 27 items in five sub-dimensions: cultural knowledge (9 items), cultural skills (6 items), cultural experience (4 items), cultural awareness (4 items), and cultural sensitivity (4 items). Questions were rated on a five-point Likert scale ranging from “not at all” (1) to “absolutely yes” (5); a higher score indicated higher cultural competence. Cronbach’s α was 0.91 in the original version [12] and 0.92 in this study.

### 2.5. Data Analysis 

The collected data were analyzed using the IBM SPSS Statistics 23 (IBM Corporation, Armonk, NY, USA). Participants” general characteristics were analyzed by real number and percentage; the level of main variables were analyzed by mean, standard deviation, minimum value, maximum value. Skewness and Kurtosis was used to identified of main variables’ normality. Differences in cultural competence according to general characteristics were analyzed using the *t*-test and analysis of variance (ANOVA), as a Post hoc after ANOVA test, Scheffé test was used. To analyze the correlations among the main variables, Pearson’s correlation coefficients were used. Further, a multiple stepwise regression analysis was performed to identify the effect factors of cultural competence, and the significance level was set to a standard of α < 0.05.

### 2.6. Ethical Consideration

This study was approved by the Institutional Review Board of Daegu Catholic University (IRB No. CUIRB-2022-0034). The investigation protocols were conducted ethically according to the World Medical Association’s Declaration of Helsinki. The participants were informed about the purpose of the experiment and gave informed consent to participate in the research. The researchers did not report any ethical, moral, or economic conflicts of interest

## 3. Results

### 3.1. General Characteristics and Differences in Cultural Competence

Of the 235 participants, 83.4% were female, with an average age of 22.5 years. Seniors made up the majority (63%), followed by freshmen (23.0%), juniors (8.0%), and sophomores (6.0%). Most participants had experience taking multi-culture classes (88.5%), 23.0% of the participants had no foreign travel experience, and 64% had contact with foreign patients. Moreover, 68.1% of the participants had no foreign friends, and most were interested in foreign news or information and preferred foreign films, dramas, and documentaries. Among the general participants’ characteristics, the variables showing a significant difference in cultural competency were age (F = 10.60, *p* < 0.001), grade (F = 5.32, *p* = 0.001), and interest in information in news or information about foreign countries (F = 8.32, *p* < 0.001), preference for foreign films, dramas, and documentaries (F = 5.72, *p* < 0.001), and experiences of contact with multicultural patients in clinical practice (t = −2.97, *p* = 0.003). Additionally, as a result of the Scheffé test, participants who were 25 years of age or older, attentive to foreign news or information, and preferred foreign films, dramas, and documentaries fairly and highly showed a high level of cultural competency (Table 1).

### 3.2. Scores of Cultural Intelligence, Ethnocentrism, and Global and Cultural Competence

The average score of cultural intelligence was 91.00 ± 17.28 (out of 140 points), ethnocentrism was 36.98 ± 7.03 (out of 75), global competence was 93.28 ± 14.78 (out of 135), and cultural competence was 95.39 ± 15.63 (out of 135 points) (Table 2).

### 3.3. Correlation among Major Variables and Cultural Competence

Cultural competence was significantly positively correlated with cultural intelligence (r = 0.59, *p* < 0.001), global competence (r = 0.62, *p* < 0.001), and significantly negatively correlated with ethnocentrism (r = −0.60, *p* < 0.015) (Table 3).

### 3.4. Factors Affecting Cultural Competence of Nursing Students 

To identify factors affecting the cultural competence of nursing students, a stepwise multiple regression analysis was performed by inputting major variables of age, grade, interest in news or information about foreign countries, preference for foreign films, dramas, documentaries, and the experience of contact with multicultural patients. A significant difference in cultural competence among participants’ general characteristics was found. First, the assumptions for regression analysis results showed the tolerance of 0.35–0.96 (≥0.1), variance inflation factor was 1.05–2.88 (≤10), and Durbin Watson value was 1.909 (approximately 2), indicating no problem of multi collinearity. The regression model was statistically significant (F = 27.21, *p* < 0.001). Factors significantly affecting the cultural competence of nursing students were cultural intelligence (β = 0.31, *p* < 0.001) and global competence (β = 0.37, *p* < 0.001). The explanatory power of these effects was 47.3% (Table 4).

## 4. Discussion

The average Cultural Competence (CC) was similar to or higher than previous studies (2.66 ± 0.62~3.38 ± 0.55 points) using the same tool before COVID-19 pandemic [8,10,19]. This is different from the results of previous studies in which the CC of nursing students and nurses was lower than before after the outbreak of COVID-19 [4,5,9,16]. This difference can be seen as the fact that direct multicultural contact opportunities have decreased under the prolonged COVID-19, but the extra time and place created by online classes and clinical practice have increased students’ chances to contact an online media content such as film, drama, information, news through YouTube, SNS; Twitter, Instagram, Facebook et al. As such, it seems that nursing students, who are digital generation, maintained and promoted CC through rapid acquisition and sharing of domestic and foreign news and information [8,11,19]. Additionally, furthermore, nursing students tend to increase CC as the grade increases [8,11,19], so it can affect the CC levels in this study which has a higher proportion of junior and senior students than previous studies (45.0~67.2%) [8,10,19]. Therefore, considering differences in academic years and the application of online education program methods can increase the effectiveness of nursing students’ cultural competence. The result of cultural competence was consistent with previous studies that cultural competence differed according to the age and grade [11,14], nursing experience of multicultural patients [8,9,11,17], and experience of acquiring multicultural information from mass-media [9]. However, there is no significant difference in CC according to age, grade, and interest in the multiculturalism of nursing students [4,8,9,10,13,14,19]. The nursing students’ cultural competence can be influenced by general characteristics including personality, environment, education, and social characteristics. Therefore, it is necessary to identify the results through repeated and expanded research in the future. Interestingly, in this study, the group with nursing experience of multicultural patients had lower CC than the group without. This is partially similar to the results of previous studies that showed that the group with ‘Rarely’ experience of multicultural patient nursing before the outbreak of COVID-19 had a lower CC level than the group with ‘Never experience’, but the ‘Frequently’ group was significantly higher than other groups [8]. Nursing students who are still in the process of becoming professional nurses, actual contact with multicultural patients can experience language barriers, insufficient nursing services provided by lack of cultural awareness and knowledge, which can reduce cultural care confidence and cultural competence. Actually, the more experience in multicultural nursing services, the higher the cultural competency [9,16]. Therefore, cultural education and practice opportunities are increased, it can be improved the cultural competencies of nursing students.

This study found that the higher the cultural intelligence and global competence level of nursing students and the lower ethnocentrism, the higher the cultural competence. Before COVID-19 pandemic, there were similar studies showing the significant positive correlation between cultural empathy and cultural competence [8,14] and cultural intelligence and cultural sensitivity of nursing students [20]. that reports the correlation between global competency and similar concepts of civic organizational behavior and CC. Additionally, positive correlations between global competence [19], global citizenship (a similar concept to global competence) [13] and cultural competence [13]. Additionally, it was partially similar to previous studies that showed ethnocentrism and cultural sensitivity of Turkish nursing students and Taiwanese university students [24,25] and a negative correlation between ethnocentrism of Korean nurses and cultural competence [16]. Since related studies are insufficient after COVID-19 pandemic, it is necessary to compare and analyze the results through future repetitive studies.

In this study, factors affecting the cultural competence of nursing students were cultural intelligence and global competence. This was partially similar to the results of previous studies that Korean nursing students’ empathy for multiculturalism positively influenced cultural competence [8,14] and cultural intelligence positively influenced to cultural sensitivity of nursing students in Turkey [16]. Additionally, global competence was an influencing factor in the cultural competency of nursing students in the Republic of Korea [19] and showing that global citizenship, a concept that recognizes and respects the diversity of values with responsibility for recognizing and solving international issues, was an influential factor in cultural competence [13]. As such, no noticeable difference was found in the results of this study compared to previous studies before COVID-19 outbreak. Cultural intelligence is a unique intelligence that helps us understand why some individuals are arguably more effective than others in an international environment [1,21,27]. Cultural education training online and offline programs can effectively increase the cultural competence of nursing students by increasing their cultural intelligence [22], help reduce the cultural difference between patients’ health and healthcare, and improve patient satisfaction and quality of health [30]. It can be improving the cultural competence and nursing students as future nursing leaders in emergency preparedness and response for global health crisis in the future. Additionally, news related to COVID-19 delivered through the mass media in real-time, the exchange of preventive vaccines, and sharing of research results for treatment in each country became an opportunity for shared responsibility for the entire global community to develop international health capabilities to protect human health. Therefore, concerning nursing students, global competence should be cultivated to increase the cultural competence required to engage in nursing professions in the Republic of Korea and abroad. Specifically, an open and positive attitude toward other cultures, the development of creative thinking to find solutions to various problems faced in the multicultural nursing fields, development of content for communication methods to enhance expression, and development and application of research and class subjects are required. Meanwhile, it was found that the cultural intelligence level was lower than that of the previous study using the same tool (95 ± 15.80) [16], meanwhile, global competence level was higher than in previous study of COVID-19(3.15 ± 0.45) [19]. Since cultural intelligence increases when there are many multicultural direct contacts [31], it aligns that the cultural intelligence of nursing students has decreased somewhat due to non-face-to-face practice and social distancing due to the influence of COVID-19. Additionally, as in the cultural competence results, the increase in opportunities for multicultural indirect contact through online seems to have grown the global competence to experience common situations under the influence of COVID-19 in real time and to participate effectively in other worlds [19,29]. 

In this study, the ethnocentrism of nursing students was not confirmed as an influencing factor of cultural competence. Although this differed from the previous study that confirmed the relationship between ethnocentrism and intercultural sensitivity of nursing students in Turkey [24], it aligned with the study that reported no significant path between the ethnocentrism of Korean clinical nurses and cultural competence [16]. Neuliep judged that the ethnocentrism was substantial when the ethnocentrism score was 55 or higher [28]. The average ethnocentrism in this study was lower than the Turkish nursing students (48.83 points) [24] measured using the same tool and the 38.60 points of clinical nurses in the Republic of Korea before COVID-19 pandemic. Additionally, it was not higher than Japanese university students (43.20 points) [23] and medical students from Iran (38.39 points) and Iraq (47.89 points) [32], indicating that the participants’ attitudes toward culture were not exclusive. The level of ethnocentrism among Korean nursing students with the prolonged experience of the COVID-19 pandemic may have decreased. In fact, COVID-19 pandemic, which started in Wuhan, China, in 2019, spread rapidly worldwide and brought a fatal health risk to the global community, at same time, anti-Chinese sentiments and stigmatization have been equally strong across Asia as in other parts of the world [33]. In the Republic of Korea, also, more than 500,000 people signed a petition requesting the government ban all Chinese visitors like other country [33]. However, each country has shared and developed COVID-19 vaccines with a shared responsibility for global health, and mutual cooperation is underway to develop treatments. Now, it is recognized once again that the world lives together as a community. Since ethnocentrism is closely related to national economic, social, and political stability and religion, differences in ethnocentrism following country characteristics and its relationships with cultural competence must be explored in future studies [32]. No study yet examines the effects of ethnocentrism and cultural competence on nursing students in the Republic of Korea during COVID-19 pandemic. 

Further research on variables with similar concepts and relationships between the variable and cultural competence is needed. Notably, the difference between previous studies’ results before the outbreak of COVID-19 pandemic and this study was not extensive. Although overseas travel and cross-border exchanges were restricted due to the impact of the COVID-19 pandemic, the opportunities for multicultural contact among college students active in various online community activities such as Facebook, YouTube, and Instagram have not de-creased. During the pandemic, offline exchanges between countries were actively conducted, which may have become an external environmental factor positively affecting the increase in cultural intelligence and global competence and lower ethnocentrism of university students; comparative studies by country are proposed in the future. Additionally, research studies and model building on the latest concepts concerning the cultural competence of nursing students and nurses in previous studies but not included in this study, such as moral sensitivity [24], intercultural willingness [32], and intercultural anxiety must be explored [16]. Lastly, since there are few qualitative studies on cultural competency targeting nursing students, these must be explored [34]. 

### Strengths and Limitations 

To our knowledge, this study investigated the relationship between cultural intelligence, ethnocentrism, global competence, and cultural competence of nursing students which had not been previously studied in the Republic of Korea. The results of this study have the following significance, which can contribute to multicultural nursing education for enhancing the cultural competence: First, the results of this study provide information that can be compared and analyzed with similar research results before the COVID-19 pandemic. Based on this, it can contribute to the development of nursing education with cultural competence that can adapt to future global health problems and provide essential nursing care service. Second, the outcomes of this study can be provided as primary data for the development of various nursing curriculum and programs to improve multicultural cultural competence of nursing students at the time of the epidemic by confirming that cultural intelligence and global competence are positive factors. At last, this study used Korean cultural competence tools to identify cultural competency and related factors. The results of this study can contribute as a basic knowledge for developing the cultural intelligence scale and an ethnocentrism scale for Korean nursing students that reflect Korean people’s cultural characteristics.

This study also has the following limitations. In this study, the recruitment of participants was done using the convenience sampling, also sampling was taken only from two institutions that was accessible by authors. Therefore, it is necessary to compare the results through repeated studies that expand the number of participants, regions, and schools, so that the results of survey studies can be generalized. The tool used in this study was developed in a foreign country except for the cultural competency tool, and it was modified and supplemented according to the Korean circumstances; however, it was challenging to accurately reflect the Korean culture. Additionally, there were difficulties in comparison and analysis because not many related studies were reported in the Republic of Korea. 

## 5. Conclusions

This study showed that cultural intelligence and global competence were identified as positive influencing factors of cultural competence of nursing students of the Republic of Korea. Although COVID-19 had a serious impact on the world, there was no significant change in the relationship between factors related to the cultural competence of Korean nursing students in this study compared to before the outbreak of COVID-19. However, our results encourage nursing students needed more cultural competence as professional nurses in the future, it is necessary to develop and apply effective education using various perspectives and methods for increasing the cultural intelligence and global competence. Using the online and face-to-face nursing education programs should include not only language, multicultural knowledge and practical health care service skill but also awareness of human rights and ethics.

## Figures and Tables

**Table 1 ijerph-19-15181-t001:** General characteristics and differences in cultural competence (N = 235).

Characteristics	Categories	N (%)	Cultural Competence (CC)	
			M ± SD	t or F(*P*) Scheffé
Age (year)	≤20 ^a^	70(29.8)	3.70 ± 0.50	10.60 (<0.001) a, b > c *
	21–24 ^b^	142(60.4)	3.53 ± 0.56	
	≥25 ^c^	23(9.8)	3.08 ± 0.67	
Gender	Female	196(83.4)	3.56 ± 0.53	−1.28 (0.209)
	Male	39(16.6)	3.40 ± 0.77	
Grades	1	54(23.0)	3.73 ± 0.47	5.32 (0.001)
	2	14(6.0)	3.73 ± 0.50	
	3	19(8.0)	3.72 ± 0.42	
	4	148(63.0)	3.42 ± 0.61	
Clinical practice experience	Yes	208(88.5)	3.56 ± 0.56	1.82 (0.071)
	No	27(11.5)	3.35 ± 0.71	
Number of trips abroad	0	54(23.0)	3.52 ± 0.58	1.01 (0.403)
	1–2	89(37.9)	3.48 ± 0.59	
	3–5	75(31.9)	3.56 ± 0.57	
	6–9	13(5.5)	3.82 ± 0.39	
	≥10	4(1.7)	3.56 ± 0.81	
Number of foreign friends	0	160(68.1)	3.48 ± 0.56	1.73 (0.144)
	1	39(16.6)	3.56 ± 0.47	
	2	20(8.5)	3.75 ± 0.80	
	4	4(1.7)	3.86 ± 0.08	
	≥5	12(5.1)	3.72 ± 0.67	
Interest in news or information about foreign countries	Not interested at all ^a^	11(4.7)	3.26 ± 0.48	8.32 (<0.001) a, b, c, d < e **
	Not very interested ^b^	56(23.8)	3.31 ± 0.49	
	Slightly interested ^c^	135(57.4)	3.58 ± 0.58	
	Moderately interested ^d^	27(11.5)	3.67 ± 0.45	
	Very interested ^e^	6(2.6)	4.49 ± 0.56	
Preference for foreign films, dramas, and documentaries	Not preferred at all ^a^	6(2.6)	3.01 ± 0.30	5.72 (<0.001) a, <d, e ***
	Not very preferred ^b^	31(13.2)	3.36 ± 0.55	
	Preferred ^c^	101(43.0)	3.46 ± 0.50	
	Significantly preferred ^d^	61(25.9)	3.62 ± 0.49	
	Most preferred ^e^	36(15.3)	3.84 ± 0.79	
Experience providing nursing care to multicultural patients	Yes	142(60.4)	3.45 ± 0.61	−2.96 (0.003)
	No	93(39.6)	3.67 ± 0.50	

SD: Standard deviation. *p* < 0.05. * As a Post hoc after ANOVA test (Scheffé) represents the significant difference of CC by each age group. ^a^, ^b^, and ^c^ mean the cc average of each age, indicating that the CC level is lower than that of other groups over the age of 25. ** e group (Very interested in news or information about foreign countries)’s CC level is higher than other groups. *** a group (Not preferred at all in foreign films, dramas, and documentaries) ‘CC level is lower than ^d, e^ group and no significant difference with b, c group.

**Table 2 ijerph-19-15181-t002:** Scores of cultural intelligence, ethnocentrism, global competence, and cultural competence (N = 235).

Variables	Range	M ± SD	Min–Max	Scale Standardization	Skewness	Kurtosis
Cultural intelligence	1–7	91.00 ± 17.28	20–140	4.55 ± 0.86	−0.20	1.83
Ethnocentrism	1–5	36.98 ± 7.04	21–63	2.46 ± 0.47	0.72	0.90
Global competence	1–5	93.28 ± 14.79	27–135	3.45 ± 0.55	−0.10	1.34
Cultural competence	1–5	95.39 ± 15.64	27–135	3.53 ± 0.58	0.03	1.13

**Table 3 ijerph-19-15181-t003:** Correlations of cultural intelligence, ethnocentrism, global competence, and cultural competence (N = 235).

Variables	Cultural Competence	Cultural Intelligence	Ethnocentrism	Global Competence
	r(*p*)	r(*p*)	r(*p*)	r(*p*)
Cultural competence	1			
Cultural intelligence	0.59 **	1		
Ethnocentrism	−0.16 *	−0.06	1	
Global competence	0.62 **	0.60 **	−0.06	1

* *p* < 0.05, ** *p* < 0.01.

**Table 4 ijerph-19-15181-t004:** Influencing factors of nursing student’s cultural competence (N = 235).

Variables	B	SE	β	t	*p*
(Constant)	41.96	10.16		4.13	<0.001
Age	−0.42	0.30	−0.09	−1.43	0.156
Academic year	−0.08	0.10	−0.01	−0.08	0.936
Interest in news or information about foreign countries	0.84	1.11	0.04	0.76	0.450
Preference for foreign films, dramas, and documentaries	1.22	0.85	0.08	1.43	0.153
Experience of contact with multicultural patients	1.50	2.26	0.05	0.66	0.508
Cultural intelligence	0.28	0.06	0.31	4.90	<0.001
Ethnocentrism	−0.20	0.11	−0.09	−1.86	0.064
Global competence	0.39	0.37	0.37	6.02	<0.001
R^2^ = 0.491, Adj R^2^ = 0.473, F = 27.21, *p* < 0.001					

Note. The unstandardized beta (B), the standard error for the unstandardized beta (SE), the standardized beta (β), the *t* test statistic (t), and the probability value (*p*). *p* < 0.05.

## Data Availability

Not applicable.

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
