# Peer review of "Factors Affecting Cultural Competence in a Sample of Nursing Students during the Prolonged COVID-19 Pandemic in Republic of Korea: A Cross-Sectional Study"

_ijerph, 2022, doi:10.3390/ijerph192215181_

Round 1
Reviewer 1 Report
The manuscript contains important information that justifies publication. The authors do not mention whether or not any reference from the EQUATOR network was used. Regarding the data collection instruments used, it would be interesting for the authors to comment on whether they were validated for the local culture and language (I believe not, but this must be described in the manuscript, with the justification). Although the discussion addresses the relationship between the data obtained in this study and the national and international literature, I felt the lack of a greater comparison of the findings with the literature. This part was somewhat limited, although the authors have commented that publications on the topic are scarce. The authors present the limitations of the study, but they could have better emphasized the contributions of their study to nursing.
Author Response
We thank you and the reviewers for your thoughtful suggestions and insights. The manuscript has benefited from these insightful suggestions. Please refer revised manuscript with response paper.

Reviewer 2 Report
It is an interesting study, written in a paper in a quite correct and descriptive way and that addresses such a necessary aspect as the development of cultural competencies in nursing students. With some important methodological limitations, which question the external validity of the study, and with some misleading factors, such as the contextualization of the study in the COVID-19 pandemic, I would like to make the following specific comments to the authors:
Abstract: It is self-sufficient and explanatory, but there are unnecessary details in a, as for example the software used or the type of online format. However, other essential methodological aspects in an abstract are missing, such as the description of the main variables and the tools used to measure these variables.
Introduction: It positions the reader well in the research problem, with a sufficient number of references related to previous studies on cultural competencies in students. However, I suggest enriching the introduction with background of qualitative studies that highlight this need for cultural competencies in professional practice among already qualified nurses and in nursing services that care for immigrants. Some suggestions:
ü Radl-Karimi C, Nielsen DS, Sodemann M, Batalden P, von Plessen C. "What it really takes" - A qualitative study of how professionals coproduce healthcare service with immigrant patients. J Migr Health. 2022 Apr 4;5:100101. doi: 10.1016/j.jmh.2022.100101. PMID: 35480876; PMCID: PMC9036136.
ü Ponce-Blandón, J.A.; Romero-Castillo, R.; Jiménez-Picón, N.; Palomo-Lara, J.C.; Castro-Méndez, A.; Pabón-Carrasco, M. Lived Experiences of African Migrants Crossing the Strait of Gibraltar to Europe: A Cross-Cultural Approach to Healthcare from a Qualitative Methodology. Int. J. Environ. Res. Public Health 2021, 18, 9379. https://doi.org/10.3390/ijerph18179379
ü Schmidt NC, Fargnoli V, Epiney M, Irion O. Barriers to reproductive health care for migrant women in Geneva: a qualitative study. Reprod Health. 2018 Mar 6;15(1):43. doi: 10.1186/s12978-018-0478-7. PMID: 29510718; PMCID: PMC5838955.
I also recommend that you review the way of citing in the introduction, since sometimes the Vancouver style is used and other times the APA style is used, and only one style should be used, according to the journal's norms.
Material and methods:
It is well described, although in the citations that are made, again different citation styles are mixed. I suggest that the ethical aspects (unformed consent, confidentiality...) be dealt with in a different subsection. Has the approval of any ethics committee been obtained?
I would just reflect that the main methodological limitations are not particularly justified: it is a descriptive cross-sectional study, with low external validity, using convenience sampling, with even more limitations in the internal and external validity of the study, and the lack of cross-cultural validation of some of the scales used in the Korean context. I suggest that proper justification be provided when describing the chosen design, the type of sampling and the scales employed.
Results:
They are correctly described, with an adequate proportion of tables and text.
Discussion:
The discussion contrasts the results of the study well with previous studies that have addressed the cultural competencies of nursing students and their associated factors. However, I miss more discussion of the elements that the pandemic may have introduced into the factors influencing cultural competencies. Personally, in fact, I do believe that the COVID-19 pandemic factor is a circumstantial issue that may contribute to skewing the results of the study, but I do not believe that it was the focus of the research problem, as seems to be implied by reading the title of the article and by reading the introduction. I suggest that this issue be revisited, as I think it is confusing.
I think that the paper has important limitations to be able to generalize its results and this is not sufficiently clarified in the limitations section. I have already described these limitations in my suggestions to the material and methods section. The authors should make an effort to recognize and justify these limitations and to better discuss the scope of the results of the study, perhaps based on previous similar studies, which have the same limitations or, lacking them, have yielded similar results.
In fact I also suggest revising the conclusions, perhaps too ambitious considering the limitations of the study. Although the suggestions to improve the cultural competencies of the curricular content of nursing studies are very laudable, I am not very sure that with the findings and limitations of the study this can be supported. I also believe that it focuses too much on quantitative studies and I believe that precisely the cultural aspects can also be approached from a qualitative perspective in an effective and complementary way to the qualitative approach.
Author Response

(The authors gave the same response as above.)

Reviewer 3 Report
Dear Authors,
I appreciate the opportunity to review this article on a very interesting topic, however, I would like to leave some suggestions for improvement and analysis.
I start with a global approach and then some more specific suggestions.
The Title suggests an approach that I haven't been able to recognize throughout the article. Did refer to the COVID-19 Pandemic in the title bring a different perspective to the article? Did the fact that the pandemic occurred, which led to the less global mobility of people in the world, made the results found different? If they consider it relevant to maintain the issue of COVID-19, I believe that they should better clarify the reason for the study and discuss the results in a deeper way, relating them to the pandemic.
Abstract:
Methods
They should mention the type of study as stated in the title: This is Cross-Sectional Study, quantitative, ...
Data collection was based on an online Google Form questionnaire.
Data collection period?
Introduction: Please justify the study and adjust the research objective to the Covid-19 issue.
Materials and Methods
2.1. Study Design: It has to be improved. It is the Cross-Sectional Study, quantitative, descriptive and correlational
2.3. Data Collection: Include that the data collection was through: an online Google Form questionnaire.
2.4. Data collection instrument
Before presenting each scale, mention that it consists of x scales that are presented below and why questionnaire for collecting sociodemographic data (the type of questions may be specified)
There is still a point for ethical issues. What is the approval number by the Ethics Committee? In addition to informed consent, were other ethical procedures followed?
Does the manuscript follow some checklist? For example STROBE checklist?
The Discussion and Conclusion list more clearly the objective of the study, during the COVID-19 pandemic and the results found, as I said above.
On study limitations: The fact that the scales are not validated for the South Korean population will have to be further emphasized as a limitation and how you controlled for possible bias.
Author Response
We thank you and the reviewers for your thoughtful suggestions and insights. The manuscript has benefited from these insightful suggestions. I'm sorry, but there are many things in common with reviewers 1 and 2, so please refer to the answers at the beginning of the document.

Round 2
Reviewer 2 Report
I really enjoyed reading the article and I must say that it has undergone a very substantial improvement, so I want to acknowledge the effort of the authors.
Some suggestions for improvement:
I think that a greater description of the statistical tests used for the analysis of relationships between variables is missing in section 2.5. For example, in the results of precibe, the use of ANOVA test (Scheffé) is not described here. The authors should make an effort to improve this section.
In tables 1 and 4, the p values that are significant should be indicated with * p<0.05 or ** p<0.01 at the foot of the table, as is done in table 3.
In the discussion, the text would be improved if some results that are cited in too much detail during the comparison with the results of other works, which have already been reproduced in the results section and need not be repeated, were eliminated.
Congratulations on the improvement of the paper
Author Response
I really thank your last comments for improving our study. All parts which were revised are in blue font in manuscript.
